# Sir2 phosphorylation through cAMP-PKA and CK2 signaling inhibits the lifespan extension activity of Sir2 in yeast

Woo Kyu Kang[1], Yeong Hyeock Kim[1], Hyun Ah Kang[2], Ki-Sun Kwon[3], Jeong-Yoon Kim[1]*

[1]Department of Microbiology and Molecular Biology, College of Bioscience and Biotechnology, Chungnam National University, Daejeon, Republic of Korea; [2]Department of Life Science, Chung-Ang University, Seoul, Republic of Korea; [3]Aging Intervention Research Center, Korea Research Institute of Bioscience and Biotechnology, Daejeon, Republic of Korea

**Abstract** Silent information regulator 2 (Sir2), an $NAD^+$-dependent protein deacetylase, has been proposed to be a longevity factor that plays important roles in dietary restriction (DR)-mediated lifespan extension. In this study, we show that the Sir2's role for DR-mediated lifespan extension depends on cAMP-PKA and casein kinase 2 (CK2) signaling in yeast. Sir2 partially represses the transcription of lifespan-associated genes, such as *PMA1* (encoding an $H^+$-ATPase) and many ribosomal protein genes, through deacetylation of Lys 16 of histone H4 in the promoter regions of these genes. This repression is relieved by Sir2 S473 phosphorylation, which is mediated by active cAMP-PKA and CK2 signaling. Moderate DR increases the replicative lifespan of wild-type yeast but has no effect on that of yeast expressing the Sir2-S473E or S473A allele, suggesting that the effect of Sir2 on DR-mediated lifespan extension is negatively regulated by S473 phosphorylation. Our results demonstrate a mechanism by which Sir2 contributes to lifespan extension.

*For correspondence:
jykim@cnu.ac.kr

**Competing interests:** The authors declare that no competing interests exist.

## Introduction

Sirtuins, a highly conserved family of nicotinamide adenine dinucleotide (NAD)-dependent protein deacetylases, have been implicated as a key metabolic sensor to link dietary restriction (DR) with lifespan extension in yeast, worms, flies, and mice (*Longo and Kennedy, 2006*). In yeast, silent information regulator 2 (Sir2), the founding member of the sirtuin family, mediates transcriptional silencing at the ribosomal DNA (rDNA) locus, mating type loci and telomeres by deacetylating the acetylated lysine 16 of histone H4 (*Rine and Herskowitz, 1987*; *Hecht et al., 1996*; *Smith and Boeke, 1997*; *Imai et al., 2000*). Sir2 is thought to affect replicative aging of yeast cells by repressing recombination and instability at the rDNA (*Sinclair and Guarente, 1997*; *Falcon and Aris, 2003*; *Ganley et al., 2009*; *Stumpferl et al., 2012*; *Kwan et al., 2013*). In addition, Sir2 may regulate lifespan by deacetylating H4K16 at sub-telomeric regions (*Dang et al., 2009*) and by asymmetrically segregating damaged proteins and cellular organelles between mother and daughter cells (*Aguilaniu et al., 2003*; *Erjavec and Nystrom, 2007*; *McFaline-Figueroa et al., 2011*). However, whether Sir2 mediates lifespan extension by DR in yeast is still under debate. Despite lots of reports supporting the role of Sir2 in DR-mediated lifespan extension (*Lin et al., 2000*, *2002*; *Anderson et al., 2003*; *Lin et al., 2004*), numerous studies challenged the hypothesis (*Jiang et al., 2002*; *Kaeberlein et al., 2004*; *Fabrizio et al., 2005*; *Smith et al., 2007*). Similarly, in worms and flies, the role of Sir2 in lifespan extension is still controversial (*Tissenbaum and Guarente, 2001*; *Rogina and Helfand, 2004*;

**eLife digest** We know that cutting calorie intake through a restricted diet can slow down the aging process and prolong the lives of many organisms ranging from yeast to mammals. Calorie restriction also has protective effects on various age-related diseases including neurodegenerative disorders, cardiovascular disease, and cancer. Many studies suggest that we may mimic the beneficial effects of calorie restriction by controlling the activities of some proteins involved in the aging process.

An enzyme called Sir2 is required for calorie restriction to be able to increase lifespan. This enzyme modifies proteins called histones, which are used to package DNA inside cells. In yeast, Sir2 modifies the histones in such a way that the genes contained in that section of DNA are inactivated (or 'silenced'). As the yeast cells age, the activity of Sir2 declines, which allows these genes to become active and contribute to the aging process. However, when yeast cells are grown in the presence of little sugar—which mimics caloric restriction—Sir2 is activated and this restores gene silencing.

It is not clear how Sir2's ability to silence these genes contributes to prolonged lifespan. Kang et al. studied the role of Sir2 in yeast and observed that one of the genes that Sir2 inactivates is called *PMA1*. This gene encodes a protein that is known to restrict the lifespan of yeast cells. Further experiments show that other proteins attach or remove molecules called phosphate groups from Sir2 to regulate its activity. Sir2 is inactivated when a phosphate group is attached, and active in the absence of phosphate. Under a reduced diet, the proteins that add phosphate to Sir2 are inactive, which allows Sir2 to become active and reduce the expression of the *PMA1* gene.

These results show that Sir2 fine-tunes the expression of *PMA1* and other age-related genes and that the attachment of phosphate groups to Sir2 by other proteins interferes with this regulation. The next challenges will be to identify the proteins responsible for attaching phosphate groups to Sir2, and to find out how they work.

*Lee et al., 2006*; *Wang and Tissenbaum, 2006*; *Burnett et al., 2011*; *Viswanathan and Guarente, 2011*; *Banerjee et al., 2012*; *Kanfi et al., 2012*).

Here, we propose a new, potentially conserved, molecular mechanism of Sir2 in DR-mediated lifespan extension. Given the potential therapeutic implications of sirtuins, understanding the complex and controversial actions of sirtuins is one of central tasks in the sirtuin biology and aging field. We demonstrate that the phosphorylation of Sir2 at S473, which is regulated through cAMP-PKA and casein kinase 2 (CK2) signaling, determines the role of Sir2 in replicative lifespan (RLS).

## Results

Deletion of *SIR2* is associated with loss of transcriptional silencing at sub-telomeric regions, sterility in haploids, destabilization of rDNA, and shortened lifespan (*Kaeberlein et al., 1999*). We observed that *sir2Δ* mutant cells were more sensitive than the wild-type (WT) strain to high concentrations of NaCl and other monovalent cations, a phenotype not previously associated with loss of Sir2 (*Figure 1A* and *Figure 1—figure supplement 1A–C*). No sensitivity to divalent cations or osmotic stress was detected in the *sir2Δ* mutant (*Figure 1—figure supplement 1A*). The NaCl sensitivity of the *sir2Δ* mutant was affected neither by Fob1, which increases the amount of extrachromosomal rDNA circles, nor by pseudodiploid state of the *sir2Δ* mutant strain (*Figure 1—figure supplement 2*).

### Sir2 negatively regulates *PMA1* transcription by deacetylating H4K16

The plasma membrane potential ($\Delta\Psi$) is a critical determinant of cation tolerance (*Navarre and Goffeau, 2000*), and we reasoned that the NaCl sensitivity of the *sir2Δ* mutant could be due to hyperpolarization of the membrane. Indeed, relative to the WT strain, the *sir2Δ* mutant was more sensitive to the $\Delta\Psi$-dependent antibiotic Hygromycin B (*Figure 1—figure supplement 1A*) and had a higher $\Delta\Psi$ (*Figure 1B*). The high $\Delta\Psi$ in the *sir2Δ* mutant could result from at least two distinct mechanisms: decreased activity of the potassium uptake system (*Madrid et al., 1998*) or increased expression of *PMA1*, which encodes a plasma membrane $H^+$-ATPase (*Serrano et al., 1986*). Reduced potassium uptake was unlikely to contribute to the membrane hyperpolarization because potassium

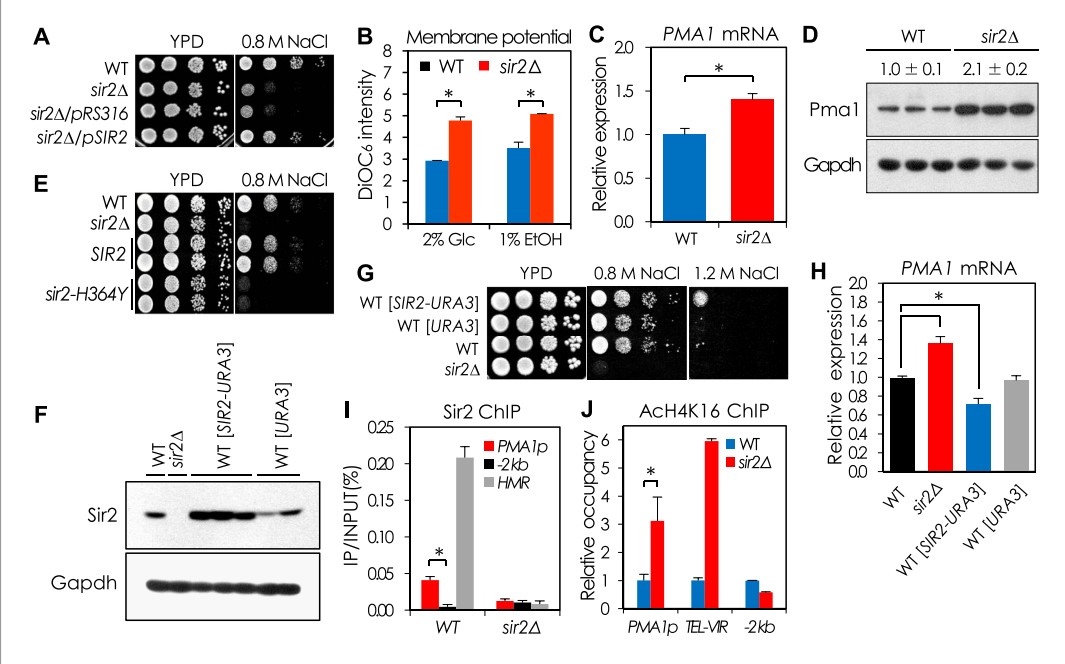

**Figure 1**. Sir2 negatively regulates *PMA1* expression by deacetylating H4K16 in the *PMA1* promoter. (**A**) NaCl sensitivity of wild-type (WT), *sir2Δ*, and *sir2Δ* cells expressing *SIR2*. (**B**) Plasma membrane potential as indicated by DiOC$_6$ staining of WT and *sir2Δ* cells grown in glucose or ethanol medium (*p < 0.005). (**C**) *PMA1* mRNA levels in WT and *sir2Δ* cells measured by qRT-PCR (*p < 0.001). (**D**) Pma1 protein levels in WT and *sir2Δ* cells measured by Western blot (WB). (**E**) NaCl sensitivity of WT, *sir2Δ*, and *sir2Δ* cells carrying WT *SIR2* or the *sir2-H364Y* mutant allele. (**F**) Silent information regulator 2 (Sir2) protein levels of the *SIR2*-overexpressing cells measured by WB. (**G**) NaCl sensitivity of the *SIR2*-overexpressing cells. (**H**) *PMA1* mRNA levels of the *SIR2*-overexpressing cells measured by qRT-PCR (*p < 0.02). (**I**) Sir2 enrichment at the *PMA1* promoter measured by Sir2 ChIP (*p < 0.001). (**J**) H4K16 acetylation levels at the *PMA1* promoter in WT and *sir2Δ* cells measured by AcH4K16 ChIP (*p < 0.02). Values in (**B**), (**C**), (**H**), (**I**), and (**J**) represent the average of at least three independent experiments (±S.D.).

The following figure supplements are available for figure 1:

**Figure supplement 1**. Sir2 is involved in regulation of monovalent cations in yeast.

**Figure supplement 2**. The NaCl sensitivity of the *sir2Δ* mutant was affected neither by Fob1 nor by pseudodiploid state.

**Figure supplement 3**. Sir2 and Sas2 antagonistically regulate *PMA1* transcription in part through regulating H4K16 acetylation at *PMA1* promoter.

chloride did not reverse the NaCl-sensitive phenotype of the *sir2Δ* mutant (*Figure 1—figure supplement 2E*). In contrast, we observed elevated levels of both *PMA1* mRNA and protein levels in *sir2Δ* cells, which could not be rescued by expression of the enzymatically inactive *sir2-H364Y* allele (*Tanny et al., 1999*) (*Figure 1C–E* and *Figure 1—figure supplement 1D,E*). Further, overexpression of Sir2 decreased NaCl sensitivity and Pma1 expression relative to WT cells (*Figure 1F–H*).

Given its role in transcriptional silencing (*Moazed, 2001*), we speculated that Sir2 might regulate *PMA1* expression by directly deacetylating H4K16 in the *PMA1* promoter region. Indeed, Sir2 binding at the *PMA1* promoter region was significantly greater than at another DNA fragment 2 kb upstream of the *PMA1* promoter, although it was not as great as at other Sir2-regulated sites, specifically *Tel-VIR* and *HMR* (*Figure 1I*). Accordingly, the H4K16 acetylation level in the *PMA1* promoter region was substantially higher in the *sir2Δ* mutant than in the WT strain (*Figure 1J*). We next tested whether Sas2, a major H4K16 acetyltransferase that antagonizes the effects of Sir2 on telomeric silencing and RLS in yeast (*Kimura et al., 2002*; *Suka et al., 2002*; *Dang et al., 2009*), opposes the Sir2-dependent

H4K16 deacetylation in the *PMA1* promoter region. A *sas2Δ* mutant showed lower Pma1 mRNA levels and less H4K16 acetylation in the *PMA1* promoter region than WT cells (*Heise and et al., 2012*), although higher *PMA1* expression and H4K16 acetylation in a *sir2Δ sas2Δ* mutant suggest additional H4K16 acetyltransferases, for example, Esa1 (*Clarke et al., 1999*; *Suka et al., 2001*; *Chang and Pillus, 2009*), replacing for Sas2 in the absence of Sir2 (*Figure 1—figure supplement 3*). Taken together, these data support the model that Sir2 plays an important role in maintaining the ΔΨ in yeast through regulation of *PMA1* expression via deacetylation of H4K16 in the *PMA1* promoter, while Sas2 antagonizes this function by acetylating H4K16 at this site.

## cAMP-PKA signaling inhibits Sir2 activity for the transcriptional repression of *PMA1* through serine 473 phosphorylation

Prior studies have shown that hyperactivation of the cyclic AMP (cAMP)-dependent protein kinase A (PKA) results in sensitivity to cellular stresses (*Stanhill et al., 1999*) and NaCl (*Figure 2—figure supplement 1*), and PKA signaling has been proposed to negatively regulate Sir2 in response to glucose availability (*Lin et al., 2000*). To examine the possibility that PKA is important for regulation of *PMA1* by Sir2, we deleted *PDE2*, encoding a high-affinity cAMP phosphodiesterase, which increased the intracellular cAMP level without affecting Sir2 expression (*Figure 2—figure supplement 1D–H*). The *pde2Δ* mutant was as sensitive to NaCl as the *sir2Δ* mutant and increased the Pma1 mRNA level as high as the *sir2Δ* mutant (*Figure 2A,B*). Furthermore, the addition of 8-Bromo-cAMP, a non-hydrolyzable cAMP analog, into the culture medium mimicked the effect of the *SIR2* deletion on *PMA1* expression (*Figure 2C*). To examine whether PKA is essential for *PMA1* regulation, we deleted *TPK1*, *TPK2*, and *TPK3*, the genes encoding the catalytic subunits of PKA, from the *pde2Δ* mutant strain. The Pma1 mRNA level in the *pde2Δ tpk1/2/3Δ* mutant strain was similar to that in WT (*Figure 2D*), therefore, we concluded that cAMP-PKA signaling controls the ability of Sir2 to regulate *PMA1* transcription.

To investigate how cAMP-PKA signaling regulates Sir2 activity in the *PMA1* promoter, we measured the H4K16 acetylation level and the amount of Sir2 bound to the *PMA1* promoter in the *pde2Δ* mutant. The Sir2 binding efficiency was unaffected, but the H4K16 acetylation level was increased in the *pde2Δ* mutant (*Figure 2E,F*), suggesting that active cAMP-PKA signaling inhibits Sir2 activity but not Sir2-binding efficiency. To investigate how cAMP-PKA signaling inhibits Sir2 activity, we analyzed the phosphorylation of Sir2 in WT, *pde2Δ*, and *pde2Δ tpk1/2/3Δ* cells. The *pde2Δ* cells showed more Sir2 phosphorylation than the WT and *pde2Δ tpk1/2/3Δ* cells, suggesting that active cAMP-PKA signaling increases Sir2 phosphorylation (*Figure 2G*).

We examined whether Sir2 S473, a conserved phosphorylation site in all known Sir2 homologs (*Gerhart-Hines et al., 2011*), is responsible for the regulation of Sir2 deacetylase activity. Mutation of the S473 residue to alanine (Sir2-S473A) abolished the Sir2 phosphorylation induced by cAMP-PKA signaling (*Figure 2—figure supplement 2*). Moreover, the Pma1 mRNA level increased in WT cells expressing the phospho-mimetic Sir2-S473E protein and decreased in *pde2Δ* cells expressing the phospho-deficient Sir2-S473A protein, matching the respective levels in WT and *pde2Δ* cells expressing WT Sir2 (*Figure 2H*). Consistently, the occupancy of H4K16 acetylation in the *PMA1* promoter region was higher in the WT cells expressing Sir2-S473E and lower in the *pde2Δ* cells expressing Sir2-S473A than that in the cells expressing WT Sir2 (*Figure 2I*). Those results suggest that Sir2 S473 is phosphorylated in response to cAMP-PKA signaling, and that phosphorylated Sir2 is unable to repress *PMA1* transcription.

## Sir2 regulates transcription of ribosomal protein genes in a cAMP-PKA-dependent manner

We observed that deletion of *SIR2* or *PDE2* resulted in cell size increase, which was not associated with rDNA destabilization or pseudodiploid state of the *sir2Δ* mutant strain (*Figure 2—figure supplement 3A,B*). Because ribosome biosynthesis rate can affect yeast cell size (*Jorgensen et al., 2004*) and Sir2 associates with actively transcribed genes (*Tsankov et al., 2006*; *Li et al., 2013*), including *PMA1* and ribosomal protein genes (RPGs) (*Tsankov et al., 2006*), we reasoned that Sir2 might regulate the expression of ribosomal proteins. The mRNA levels of many RPGs were higher in *sir2Δ* and *pde2Δ* cells than in WT cells (*Figure 2—figure supplement 3C*). Because the amount of Sir2 bound to the *RPL3* and *RPL5* promoters was not changed in the *pde2Δ* mutant (*Figure 2—figure supplement 3D*),

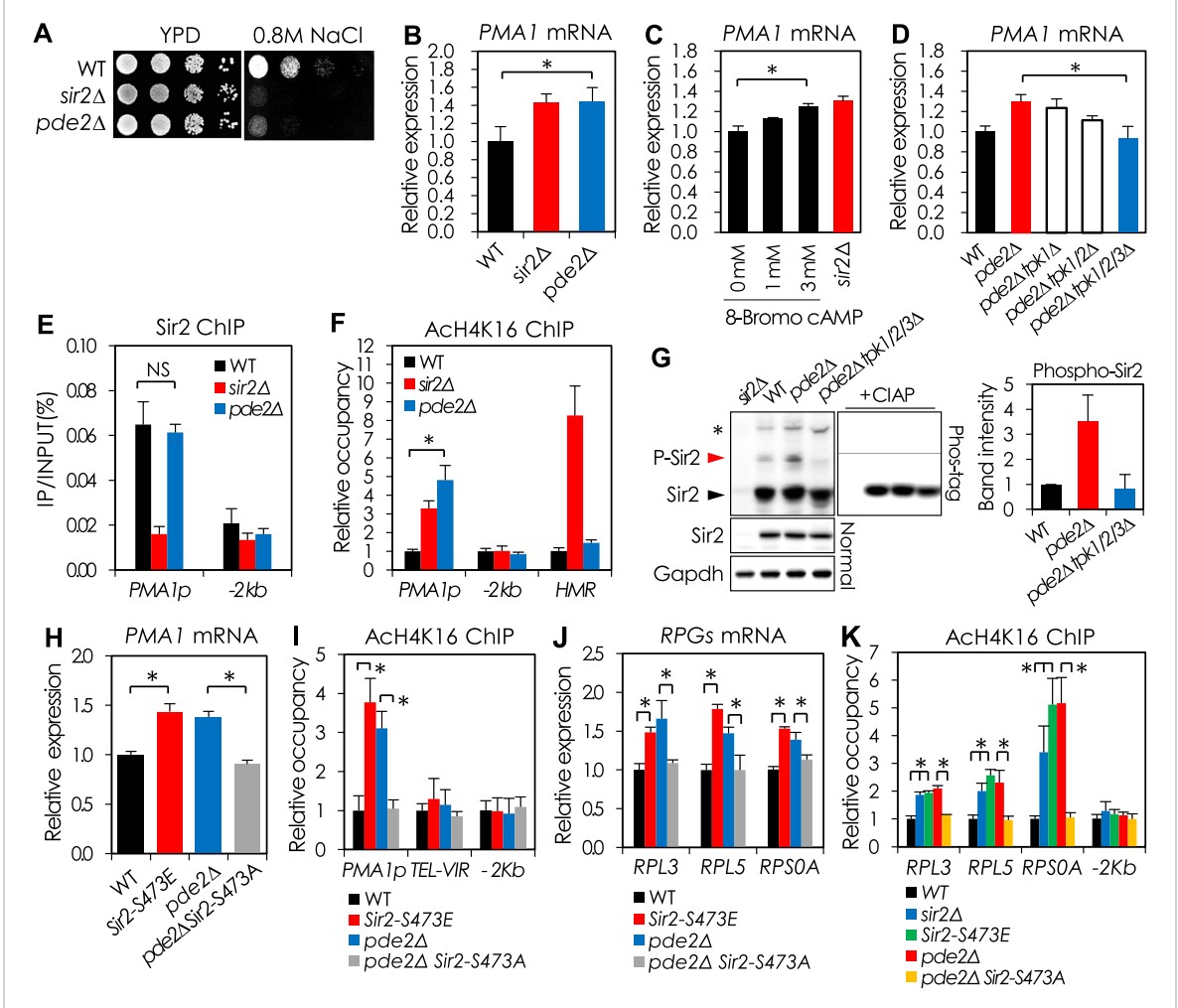

**Figure 2**. cAMP-PKA signaling inhibits Sir2 activity for the transcriptional repression of *PMA1* and *RPGs* through serine 473 phosphorylation. (**A**) Effects of *PDE2* deletion on NaCl sensitivity. (**B**) Effects of *PDE2* deletion on *PMA1* expression measured by qRT-PCR (*p < 0.01). (**C**) Effects of 8-Bromo-cAMP addition on *PMA1* expression measured by qRT-PCR (*p < 0.01). (**D**) Effects of *TPK* deletion on *PMA1* expression in *pde2Δ* cells measured by qRT-PCR (*p < 0.01). (**E**) Sir2 enrichment at the *PMA1* promoter in WT and *pde2Δ* cells measured by Sir2 ChIP (NS, not significant). (**F**) H4K16 acetylation levels at the *PMA1* promoter in WT, *sir2Δ*, and *pde2Δ* cells measured by AcH4K16 ChIP (*p < 0.001). (**G**) Sir2 phosphorylation levels in WT, *pde2Δ*, and *pde2Δ tpk1/2/3Δ* cells analyzed by Phos-tag SDS-PAGE and WB. Arrowheads indicate cAMP-PKA-dependent phosphorylated (red) and non-phosphorylated (black) Sir2. The asterisk indicates cAMP-PKA-independent phosphorylation of Sir2. (**H**) Effects of *SIR2-S473E* or *SIR2-S473A* on *PMA1* expression in WT and *pde2Δ* cells measured by qRT-PCR (*p < 0.005). (**I**) Effects of *SIR2-S473E* or *SIR2-S473A* on H4K16 acetylation at the *PMA1* promoter in WT and *pde2Δ* cells measured by AcH4K16 ChIP (*p < 0.05). (**J**) Effects of *SIR2-S473E* or *SIR2-S473A* on the expression of ribosomal protein genes (*RPGs*) in WT and *pde2Δ* cells measured by qRT-PCR (*p < 0.005). (**K**) Effects of *SIR2-S473E* or *SIR2-S473A* on H4K16 acetylation at the *RPG* promoters in WT and *pde2Δ* cells measured by AcH4K16 ChIP (*p < 0.05). Values in (**B**), (**C**), (**D**), (**E**), (**F**), (**H**), (**I**), (**J**), and (**K**) represent the average of at least three independent experiments (±S.D.).

The following figure supplements are available for figure 2:

**Figure supplement 1**. The cAMP-PKA signaling-dependent effects of SIR2 deletion on NaCl sensitivity and *PMA1* expression.

**Figure supplement 2**. Effect of sir2-S473A mutation on the phosphorylation of Sir2 in WT and pde2Δ cells.

**Figure supplement 3**. cAMP-PKA signaling and phosphorylation of Sir2 at S473 regulate the expression of genes encoding ribosomal subunit proteins and cell size homeostasis.

we thought that the cAMP-PKA-dependent phosphorylation of Sir2 is also responsible for the repression of RPGs transcription as shown in the *PMA1* regulation. WT cells expressing Sir2-S473E produced more RPG (*RPL3*, *RPL5*, and *RPS0A*) mRNAs than WT cells expressing WT Sir2, and *pde2Δ* cells expressing Sir2-S473A produced fewer RPG mRNAs than *pde2Δ* cells expressing WT Sir2 (*Figure 2J*). Consistent with the increased amounts of RPG mRNAs, the relative occupancy of H4K16 acetylation in the RPG promoters was higher in the *sir2Δ*, *pde2Δ*, and *SIR2-S473E* cells than in the WT cells (*Figure 2K*). The expression of Sir2-S473E in WT cells, or that of Sir2-S473A in *pde2Δ* cells, reversed the cell size of each strain (*Figure 2—figure supplement 3E*). These observations suggest that Sir2 regulates the transcription of many RPGs in a cAMP-PKA-dependent manner.

### *CKA2* mediates cAMP-PKA-dependent Sir2 phosphorylation

Because the amino acid residues flanking Sir2 S473 do not constitute a PKA consensus motif, other kinases downstream of PKA may be involved in S473 phosphorylation. We screened a collection of yeast kinase–gene deletions consisting of 121 mutant strains (*Figure 3—source data 1*) and selected 21 kinase mutants that grew better than WT cells in medium containing 0.8 M NaCl. Among these, we focused on *CKA2*, *KSS1*, and *DBF2*, because those three kinases are known to localize in the nucleus and the sensitivity of each of the corresponding deletion mutants to NaCl was increased in the presence of nicotinamide, a Sir2 inhibitor (*Figure 3—figure supplement 1A,B*). Only the *CKA2* deletion decreased NaCl sensitivity and the *PMA1* and *RPL3* mRNA levels of the *pde2Δ* mutant (*Figure 3A–D* and *Figure 3—figure supplements 1C–F, 2*), and the phospho-mimetic Sir2-S473E protein increased NaCl sensitivity, cell size, and *PMA1* and *RPL3* mRNA levels of the *cka2Δ* mutant (*Figure 3E–G*). The *CKA2* deletion reduced the phosphorylation of Sir2, which was significantly increased in the *pde2Δ* mutant (*Figure 3H*), and the increased interaction of Cka2 with Sir2 in the *pde2Δ* mutant was reversed in the absence of the *TPK* genes (*Figure 3I*). Furthermore, chromatin immunoprecipitation analysis revealed that Cka2 binding to the *PMA1* and *RPL3* promoter regions, but not to other Sir2 target sites including *Tel-VIR* and *HMR*, was increased in the *pde2Δ* mutant but decreased in the *pde2Δ tpk1/2/3Δ* mutant (*Figure 3J*). Collectively, those results support the hypothesis that Cka2 works downstream of PKA to phosphorylate Sir2 bound to the *PMA1* promoter region.

### Sir2 S473 phosphorylation inhibits DR-mediated lifespan extension by Sir2

Since DR is known to reduce cAMP-PKA activity (*Lin et al., 2000*) and both *PMA1* and RPGs are critical regulators of yeast RLS (*Steffen et al., 2008*; *Ito et al., 2010*; *Henderson et al., 2014*), we investigated the relevance of Sir2 S473 phosphorylation in the DR-mediated lifespan extension. The RLS in 2% glucose medium of the *SIR2-S473A* and *SIR2-S473E* cells was about 20% longer and about 10% shorter, respectively, than that of the WT cells (*Figure 4A*). The short RLS of the *pde2Δ* cells was increased by the *SIR2-S473A* allele to equal that of the WT cells in 2% glucose medium (*Figure 4B*). The *SIR2-S473A* allele had no effect on the RLS in 0.5% glucose medium (*Figure 4C*), however, which is in agreement with the *PMA1* and *RPL3* mRNA levels (*Figure 4D,E*). In addition, the effect of Sir2 S473 phosphorylation on RLS of yeast cells was independent of rDNA recombination/stability (*Figure 4F,G*). Collectively, those results suggest that the phosphorylation of Sir2 S473 inhibits DR-mediated lifespan extension by Sir2.

## Discussion

In this study, we propose a new, potentially conserved, molecular mechanism of Sir2 in DR-mediated lifespan extension (*Figure 4H*). We show that Sir2 is able to increase RLS of yeast cells only under conditions where cAMP-PKA and CK2 signaling is not active. This result suggests that the role of Sir2 in DR-mediated lifespan extension depends on the metabolic status of cells, which is also supported by our previous report that the role of Sir2 in mediating oxidative stress resistance and chronological lifespan is growth-phase dependent (*Kang et al., 2014*). We speculate that a similar paradigm may exist in higher eukaryotes, although the functions and molecular mechanisms of sirtuins are much more complex, because Sir2 S473 is a conserved phosphorylation site in all known sirtuins (*Gerhart-Hines et al., 2011*) and numerous papers report the genetic and molecular interaction between cAMP-PKA and CK2 and SirT1 phosphorylation in mammalian cells (*Kang et al., 2009*; *Zschoernig and Mahlknecht, 2009*; *Gerhart-Hines et al., 2011*; *Dixit et al., 2012*; *Park et al., 2012*; *Lee et al., 2014*).

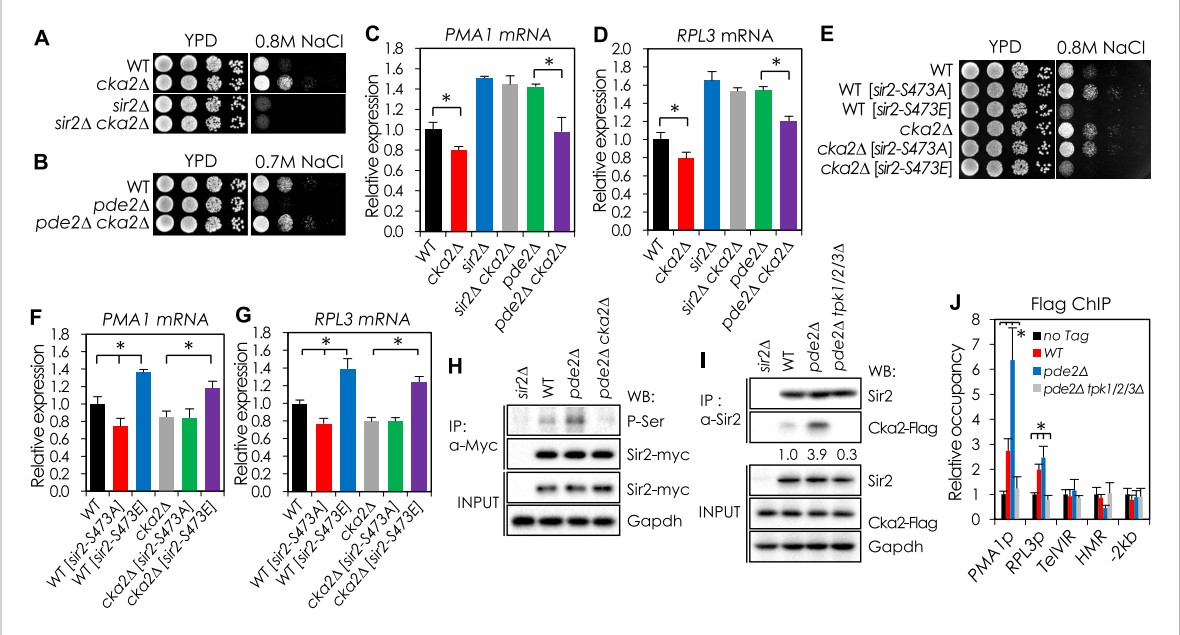

**Figure 3**. *CKA2* mediates cAMP-PKA-dependent Sir2 phosphorylation to regulate the expression of *PMA1* and RPGs. (**A**, **B**) Effects of *CKA2* deletion on NaCl sensitivity in WT (**A**), *sir2Δ* (**A**), and *pde2Δ* cells (**B**). (**C**, **D**) Effects of *CKA2* deletion on the expression of *PMA1* (**C**) and *RPL3* (**D**) in WT, *sir2Δ*, and *pde2Δ* cells measured by qRT-PCR (*p < 0.01). (**E**) Effects of *SIR2-S473E* or *SIR2-S473A* on NaCl sensitivity in WT and *cka2Δ* cells. (**F**, **G**) Effects of *SIR2-S473E* or *SIR2-S473A* on the expression of *PMA1* (**F**) and *RPL3* (**G**) in WT and *cka2Δ* cells measured by qRT-PCR (*p < 0.005). (**H**) Sir2 phosphorylation levels in WT, *pde2Δ*, and *pde2Δ cka2Δ* cells. Myc-tagged Sir2 proteins were immunoprecipitated (IP) and analyzed by WB as indicated. (**I**) In vivo Sir2 and Cka2 interaction in WT, *pde2Δ*, and *pde2Δ tpk1/2/3Δ* cells. Flag-tagged Cka2 proteins (Cka2-Flag) were IP and analyzed by WB. (**J**) Cka2-Flag enrichment at the *PMA1* promoter in WT, *pde2Δ*, and *pde2Δ tpk1/2/3Δ* cells measured by Flag ChIP (*p < 0.001). Values in (**C**), (**D**), (**F**), (**G**), and (**J**) represent the average of at least three independent experiments (±S.D.). NaCl sensitivity of the 121 kinase mutant strains used to identify kinases required for protein kinase A (PKA)-dependent Sir2 phosphorylation is available in the *Figure 3—source data 1*.

The following source data and figure supplements are available for figure 3:

**Source data 1**. NaCl sensitivity of the kinase mutant strains.

**Figure supplement 1**. Sir2-mediated NaCl sensitivity of kinase mutant strains.

**Figure supplement 2**. Effect of cka2Δ or sir2-S473E mutation on cell size of WT and pde2Δ mutant cells.

We also demonstrate that Sir2 fine-tunes transcription of the *PMA1* and RPGs by deacetylating H4K16 in the promoter. This finding indicates that Sir2-dependent H4K16 deacetylation plays a role in the regulation of actively transcribed genes in addition to the silencing of the rDNA locus, mating type loci, and telomeres in yeast. Considering previous studies showing that Sir2 binds on actively transcribed genes including *PMA1* and RPGs in yeast (*Tsankov et al., 2006*; *Li et al., 2013*), we expect this result will open a new perspective about the molecular functions of Sir2.

Pma1 protein level increases almost twofold in the absence of Sir2 (*Figure 1D* and *Figure 1—figure supplement 1E*). Recent reports indicate that vacuolar acidity is functionally linked with mitochondria and autophagy, which have a central role in the aging process (*Hughes and Gottschling, 2012*; *Ruckenstuhl et al., 2014*). And, vacuolar acidity declines in aging yeast cells because of Pma1 accumulation that reduces cytosolic protons (*Henderson et al., 2014*). It is thought that Pma1 expression affects yeast cell aging by changing intracellular pH, vacuolar pH, amino acid import into vacuole, and mitochondrial function (*Hughes and Gottschling, 2012*; *Henderson et al., 2014*). Thus, our study suggests a possibility that Sir2 contributes to DR-mediated lifespan extension at least in part by affecting mitochondrial function through cytoplasmic and vacuolar pH regulation in a context-dependent manner.

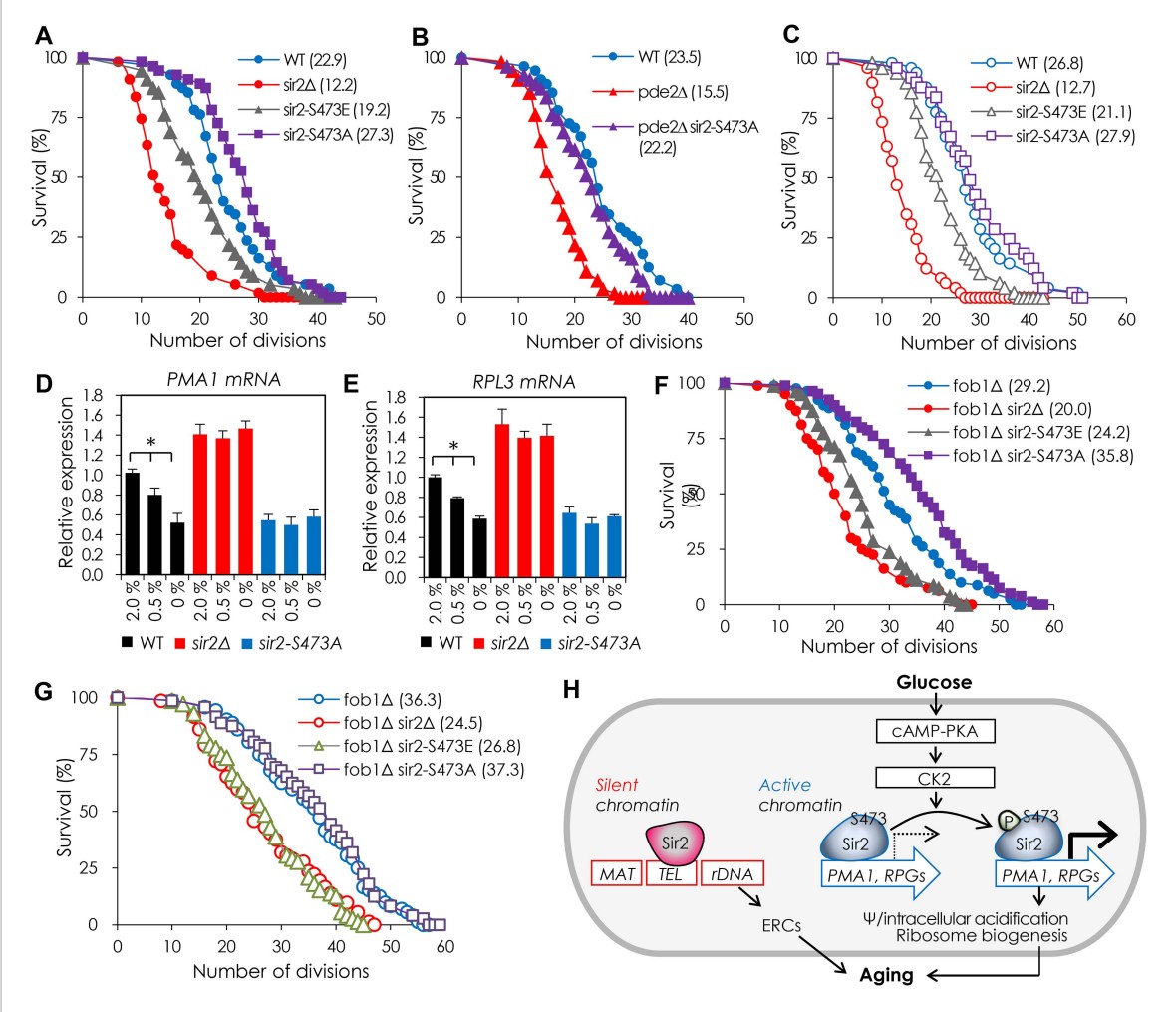

**Figure 4**. Sir2 S473 phosphorylation inhibits DR-mediated lifespan extension by Sir2. (**A**) Replicative lifespan (RLS) of the strains expressing *SIR2-S473E* or *SIR2-S473A* measured by micromanipulation. The median lifespan is indicated. p < 0.0001 (WT vs *sir2Δ*), p = 0.0013 (WT vs *sir2-S473E*), p = 0.0062 (WT vs *sir2-S473A*). (**B**) Effect of *SIR2-S473A* on the RLS of *pde2Δ* cells. The median lifespan is indicated. p < 0.0001 (WT vs *pde2Δ*), p < 0.0001 (*pde2Δ* vs *pde2Δ sir2-S473A*). (**C**) RLS of WT, *sir2Δ*, and strains expressing *SIR2-S473E* or *SIR2-S473A* grown under 0.5% glucose conditions. The median lifespan is indicated. p < 0.0001 (WT vs *sir2Δ*), p = 0.0006 (WT vs *sir2-S473E*), p = 0.5051 (WT vs *sir2-S473A*). (**D**, **E**) Effects of *SIR2-S473A* on the expression of *PMA1* (**D**) and *RPL3* (**E**) in cells grown under 2.0%, 0.5%, or 0% glucose conditions measured by qRT-PCR (*p < 0.005). The values represent the average of at least three independent experiments (±S.D.). (**F**, **G**) Effects of *SIR2-S473E* or *SIR2-S473A* on the RLS of *fob1Δ* background under 2% (**F**) and 0.5% (**G**) glucose conditions. The median lifespan is indicated. p < 0.0001 (*fob1Δ* vs *sir2Δ* under 2.0% glucose), p < 0.0001 (*fob1Δ* vs *sir2-S473E* under 2.0% glucose), p = 0.0002 (*fob1Δ* vs *sir2-S473A* under 2.0% glucose), p < 0.0001 (*fob1Δ* vs *sir2Δ* under 0.5% glucose), p < 0.0001 (*fob1Δ* vs *sir2-S473E* under 0.5% glucose), p = 0.3221 (*fob1Δ* vs *sir2-S473A* under 0.5% glucose). (**H**) A working model for how Sir2 regulates dietary restriction (DR)-mediated lifespan in yeast.

## Materials and methods

### Yeast strains and growth conditions

The yeast strains used in the study are listed in *Supplementary file 1*. The experiments were performed using the BY4741 strain, unless otherwise noted. The 10560-2B strain was used to compare and confirm the results obtained with the BY4741 strain. Deletion strains were generated by replacing each open reading frame with *URA3* through homologous recombination. To confirm the effects of *SIR2* mutations on stress resistance, a centromeric plasmid (pRS316) containing the *SIR2* promoter (−1000 to −1), the entire *SIR2* gene, and the *ADH1* terminator was introduced into a *sir2Δ* mutant strain. To generate the *SIR2* overexpression strains, a fragment that included a *SIR2* promoter, an

entire *SIR2* gene, an *ADH1* terminator, and *URA3* was integrated at the *URA3* locus of WT or *pde2Δ* strains by homologous recombination. To generate strains expressing a Sir2 protein with no deacetylase activity (sir2-H364Y) or a mutated phosphorylation site (sir2-S473E or sir2-S473A), fragments that included a *SIR2* promoter, each mutated *SIR2* gene, an *ADH1* terminator, and *URA3* were integrated at the endogenous *SIR2* promoter locus in the *sir2Δ*, *pde2Δ sir2Δ*, or *cka2Δ sir2Δ* mutant strains by homologous recombination. To facilitate Western blotting and immunoprecipitation experiments, endogenous *SIR2*, *PMA1*, and *CKA2* were tagged at the C-terminus with *13 MYC-URA3* or *FLAG-URA3* fragment by homologous recombination. All strains generated in this study were verified using PCR and/or Western blotting.

Yeast cells were routinely grown in YPD (1% yeast extract, 2% peptone, and 2% glucose) at 30℃. Synthetic complete medium lacking uracil was used for the selection of $URA^+$ strains. When required, transformants were plated onto solid medium containing 5′-fluoroorotic acid (1 mg/ml) to select for the loss of the *URA3* marker.

### Stress resistance test

Yeast cells were grown in YPD medium at 30℃ for 1 day and then seeded into 25–50 ml YPD medium at an initial $OD_{600} = 0.2$ and incubated to log phase. One milliliter of the cell culture was collected, and the cells were washed with distilled water, diluted, and spotted onto regular YPD or YPD containing various concentrations of chemicals (for cationic stress: 0.15–1.2 M NaCl, 0.2–0.8 M KCl, 0.5 M LiCl, 50 mM CsCl, 5 mM–1 M $CaCl_2$, or 300 μg/ml Hygromycin B; for osmotic stress: 1 M sorbitol or 1 M mannitol; for acidic stress: 50 mM citrate buffer [pH 3.5]). The cells were incubated for 2–3 days at 30℃, and the plates were photographed.

### Measurement of plasma membrane potential

The relative plasma membrane potential of each strain was measured as described previously (*Madrid et al., 1998*). Briefly, cells were grown in YPD ($OD_{600} < 0.5$), harvested, resuspended to $OD_{600} = 0.1$ in PBS (Phosphate-buffered saline) buffer, and exposed to 1 nM $DiOC_6(3)$ cyanine dye (3,3′-dihexyloxacarbocynine iodide, Molecular Probes, Eugene, OR) for 30 min at 30℃ in the dark. The fluorescence values were calculated by flow cytometer (Becton Dickinson, San Jose, CA). All measurements were made at least three times using independent preparations.

### Preparation of whole-cell extracts and Western blotting

Total cell extracts were prepared using the TCA method (*Keogh et al., 2006*). Proteins were separated on 8–15% SDS-PAGE gels and transferred to polyvinylidene fluoride (PVDF) membranes (Millipore, Billerica, MA). The membranes were probed with specific antibodies, and immuno-reactivity was detected using enhanced chemiluminescence reagent (Elpis Biotech, Korea). The primary antibodies were anti-Sir2 (1:200, Santa Cruz, Dallas, TX), anti-FLAG (1:1000, Sigma, Saint Louis, MO), anti-Myc (1:1000, Santa Cruz), anti-GAPDH (1:10,000, Acris, Germany), anti-AcH4K16 (1:2000, Upstate, Lake Placid, NY), and anti-H4 (1:0000, Millipore). Band density trace and quantification were determined using ImageJ (National Institutes of Health).

### Phos-tag SDS-PAGE and immunoprecipitation

To detect phosphorylated Sir2 proteins, total cell extracts were separated on 6% SDS-PAGE gels containing 25 mM Phos-tag (Wako, Japan) and 100 mM $MnCl_2$ as recommended by the supplier and analyzed by Western blot (WB) with anti-Sir2 antibody (1:200, Santa Cruz). Band intensities were quantified using ImageJ software (National Institutes of Health). Sir2 phosphorylation levels were calculated by subtracting the up-shifted Sir2 protein levels detected by Phos-tag SDS-PAGE from the total Sir2 protein levels of the same sample detected by normal SDS-PAGE. Alternatively, Sir2-13Myc proteins were immunoprecipitated (IP) using an anti-Myc antibody (1:100, Santa Cruz) and then analyzed by WB with anti-Phosphoserine antibody (1:100, Qiagen, Valencia, CA).

To investigate the interaction between Sir2 and Cka2 in vivo, cells expressing Flag-tagged Cka2 were resuspended in lysis buffer (50 mM HEPES pH 7.5, 140 mM NaCl, 1 mM EDTA, 1% Triton-100, 1 mM PMSF, 1 mM $Na_3VO_4$, and 1 mM NaF) and lysed using glass beads with vigorous vortexing. The lysates were IP overnight using anti-Sir2 antibody (1: 100, Santa Cruz) and 20 μl Protein A/G agarose

beads (Santa Cruz). The beads were then washed five times with lysis buffer, and the proteins bound to the beads were analyzed by WB with anti-FLAG antibody (1:1000, Sigma).

## cAMP extraction and determination

The method for cAMP extraction using TCA was modified from a protocol for ATP extraction (*Gustafsson, 1979*). Briefly, $2 \times 10^8$ cells (20 ml culture with $OD_{600} = 0.5$) were pelleted, washed, and resuspended in 1 ml cold milliQ-water. Metabolites were extracted by adding 1.2 ml TCA (0.5 M) and vigorously vortexing while the samples were kept on ice for 15 min. TCA was removed by ether extraction. The cAMP levels in the extracts were determined using cAMP Direct Immunoassay Kit (Biovision, San Francisco, CA) as recommended by the supplier. The values were normalized to dry cell weight.

## RNA isolation, cDNA synthesis, and real-time PCR analysis

Total RNA was purified with the RNeasy Mini kit (Qiagen) and quantified by measuring the absorbance at 260 nm. From each 0.5 µg RNA sample, cDNA was synthesized using First Strand cDNA synthesis kit (Invitrogen, Carlsbad, CA) according to the manufacturer's recommendations and analyzed by quantitative RT-PCR with the oligonucleotides described in *Supplementary file 2*. Real-time PCR was performed with SYBR green PCR mix (Bio-Rad) and CFX connect system (Bio-Rad, Hercules, CA). Relative expression levels (normalized to *ACT1*) were determined using the comparative CT method.

## Chromatin immunoprecipitation

All chromatin immunoprecipitation (ChIP) assays were performed at least in triplicate using independent chromatin preparations. The ChIP assay was carried out essentially as described (*Nelson et al., 2006*). Briefly, cells were fixed by 1% formaldehyde (Sigma) for 1 hr and quenched in 125 mM glycine. Cells were harvested and lysed using glass beads with vigorous vortexing, and the lysates were sonicated at 4°C for 10 cycles of 1 min on and 1 min off. Then, the supernatant was IP with anti-Sir2 antibody (Santa Cruz), anti-AcH4K16 antibody (Abcam, Cambridge, MA), or anti-FLAG antibody (Sigma) followed by incubation with BSA-coated Protein A/G agarose beads (Santa Cruz). The crosslinks were reversed, and IP DNA was precipitated and purified. Quantitative real-time PCR was performed to amplify specific regions using each oligonucleotide described in *Supplementary file 2*.

## Cell size measurement

Cells were harvested at an $OD_{600}$ ~0.5, and the average area of the unbudded single cells was measured from microscopic images of approximately 150 cells per sample using ImageJ software (National Institutes of Health).

## Screening of the yeast kinase–gene deletion collection

To identify kinases required for PKA-dependent Sir2 phosphorylation, we screened 121 mutant strains harboring kinase deletions in the BY4741 background for resistance to NaCl. The strains were first grown in 100 µl YPD containing 150 µg/ml G418 (Gibco-BRL, Rockville, MD) in 96-well plates at 30°C with vigorous shaking (240 rpm) for 1 day and then inoculated into 50 µl regular YPD or YPD containing 0.8 M NaCl in 96-well plates using a 96-pin replicator (V&P Scientific, San Diego, CA). The plates were incubated at 30°C with vigorous shaking (240 rpm), and the $OD_{600}$ of each well was read every 1 hr for 9 hr using a microplate reader (Bio-Rad). The NaCl resistance of each strain was calculated on the basis of growth in YPD containing 0.8 M NaCl relative to that in regular YPD (*Figure 3—source data 1*). The NaCl-resistant mutants identified during the screening were further tested for Sir2-dependence of the NaCl resistance using a spotting assay with serial dilution on plates containing 5 mM nicotinamide (Sigma).

## Yeast lifespan determination

The RLSs of the yeast strains were determined by micromanipulation as previously described (*Kaeberlein et al., 2005*) using 50–100 virgin cells grown on standard YPD plates containing 2% or

0.5% glucose. Statistical significance of the difference in the RLS between strains was determined by a two-tailed Wilcoxon rank-sum test ($p < 0.05$).

## Acknowledgements

We thank Matt Kaeberlein at University of Washington, Brian Kennedy at Buck Institute, and Joon Ho Lee and Cheon Ah Kim at Seoul National University for critical reading of the manuscript and valuable comments and to Won Ki Hur at Seoul National University for providing yeast deletion library. We are grateful to Ji Young Lee and Young Eun Kim for experimental assistance. This research was supported by grants from Korean Research Foundation (2010-0013086) and the Bio & Medical Technology Development Program (2013M3A9B6076413) of National Research Foundation funded by the Korean government (MSIP).

## Additional information

### Funding

| Funder | Grant reference | Author |
|---|---|---|
| National Research Foundation of Korea | 2010-0013086 | Jeong-Yoon Kim, Ki-Sun Kwon |

The funder had no role in study design, data collection and interpretation, or the decision to submit the work for publication.

### Author contributions

WKK, Conception and design, Acquisition of data, Analysis and interpretation of data, Drafting or revising the article; YHK, Acquisition of data, Analysis and interpretation of data; HAK, K-SK, J-YK, Conception and design, Analysis and interpretation of data, Drafting or revising the article

## Additional files

### Supplementary files

• Supplementary file 1. Strains used in the study.

• Supplementary file 2. Primers used in the study.

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
