## [Decision Letter]

Thank you for submitting your work entitled “cAMP-PKA and CK2 Signaling Determines a Novel Lifespan Extension Activity of Sir2 in Yeast” for peer review at *eLife*. Your submission has been favorably evaluated by Jim Kadonaga (Senior Editor) and three reviewers, one of whom is a member of our Board of Reviewing Editors.

The reviewers have discussed the reviews with one another, and the Reviewing editor has drafted this decision to help you prepare a revised submission. All reviewers felt that the work was interesting, important and thorough. However, there were multiple technical points that would need to be addressed prior to publication in *eLife*.

Essential revisions:

1) Despite the title, the only experiment on life span extension, in the last figure, is wrongly interpreted. The lifespan data show that phosphorylation of Sir2 negatively regulates the lifespan extension activity of Sir2. This is in contrast to the title, Abstract, Results and Discussion, which interpret this result to indicate that the phosphorylation promotes lifespan extension. The text needs to be changed to appropriately interpret the data.

2) *PMA1* expression and H4K16 acetylation was affected by deleting *PDE2*, the high-affinity cAMP phosphodiesterase. This mutant showed hyperphosphorylated Sir2 in Figure 2, and this was speculated to alter Sir2 deacetylation activity. If this were the case, then it seems as though Sir2 activity at *HMR* would have been impacted, which it was not. You need to explain your ideas as to why *HMR* silencing was not changed.

3) The data shown in Figure 1—figure supplement 3, that over expression of SIR2 causes resistance to salt and decreases *PMA1* expression, is an important finding and should be included in the main body of the text/figures.

4) The data regarding Sas2 and Sir2 do not fit your model and are inconsistent amongst themselves. Figure 1–figure supplement 4D shows that the *sir2∆ sas2∆* double mutant has less H4 K16ac, just like the *sas2∆*. This makes sense. However, the *sir2∆ sas2∆* double mutant has higher H4K16Ac by ChIP and has higher *PMA1* expression. The ChIP data suggest additional HATs for H4K16 at these loci; however the WB suggest that Sas2 is responsible for most of H4K16ac inside the cell. You need to explain what is a likely explanation for this contradiction.

5) Your proposal that *PMA1* and ribosomal protein gene regulation by Sir2 is a novel mechanism of regulating replicative lifespan and mediating DR effects, would be greatly strengthened by testing whether the *sir2-S473* mutants alter ribosomal DNA recombination/stability. Ideally, you should combine these mutants with a deletion of *FOB1* to stabilize the rDNA and then test if there is still an effect on lifespan.

6) Does the *S473A* mutant still interact with Cka2-flag through co-IP assays (as in Figure 3)? According to the triple *TPK* mutant result, it shouldn't if phosphorylation at this site is required for the interaction. If it does, there could be other important phosphorylation sites involved. Similarly, they should show that the *S473A* mutant no longer shows the phosphorylation shift on the gel.

7) The Results and Discussion section is actually all results without any substantial discussion. At a minimum, you need to add an explanation of the proposed model in Figure 4, and relate the new findings and model to other models of Sir2-mediated gene regulation or lifespan effects.

8) The actual *P* values for each yeast replicative lifespan assays need to be specifically stated in the figure legends or in a separate table.

9) For all real-time PCR data (RNA and ChIP), the number of biological replicates *(n)* performed needs to be indicated, to allow the reader to assess the quality and consistency of the data.

10) In Figure 1—figure supplement 1, panel B and C, the NaCl sensitivity by *sir2∆* seems much less than all the other NaCl or monovalent cation sensitivity experiments shown (Figure 1; Figure 1—figure supplement 1;, Figure 1—figure supplement 2; Figure 1—figure supplement 3; Figure 1–figure supplement 4A; Figure 2; Figure 2—figure supplement 1 and Figure 3). What is the cause of inconsistency? Are there other biological replicate experiments showing better consistency?

11) In Figure 1–figure supplement 4, panel A, the *sas2∆* strain grew poorly on YPD, which is even worse than in NaCl. Is this unexpected result reproducible?

---

## [Author Response]

*1) Despite the title, the only experiment on life span extension, in the last figure, is wrongly interpreted. The lifespan data show that phosphorylation of Sir2 negatively regulates the lifespan extension activity of Sir2. This is in contrast to the title, Abstract, Results and Discussion, which interpret this result to indicate that the phosphorylation promotes lifespan extension. The text needs to be changed to appropriately interpret the data*.

To avoid any misunderstanding, we changed the title to: “Sir2 phosphorylation through cAMP-PKA and CK2 signaling inhibits the lifespan extension activity of Sir2 in Yeast”. We also rephrased sentences that did not clearly state the negative role of Sir2 S473 phosphorylation in lifespan extension activity of Sir2 (e.g. Abstract, Introduction and subsection “Sir2 S473 phosphorylation inhibits DR-mediated lifespan extension by Sir2”).

*2)* PMA1 *expression and H4K16 acetylation was affected by deleting* PDE2*, the high-affinity cAMP phosphodiesterase. This mutant showed hyperphosphorylated Sir2 in*
Figure 2*, and this was speculated to alter Sir2 deacetylation activity. If this were the case, then it seems as though Sir2 activity at* HMR *would have been impacted, which it was not. You need to explain your ideas as to why* HMR *silencing was not changed*.

As the reviewers point out, our data show that cAMP/PKA signaling inhibits the deacetylase activity of Sir2 at *PMA1* and *RPGs*, but not at *HMR* (Figure 2), which suggests an intriguing hypothesis that the effect of Sir2 phosphorylation through cAMP/PKA signaling may be site-specific. We show that Cka2 phosphorylates Sir2 only at the *PMA1* and *RPL3*, not at other sites including *HMR,* in a cAMP/PKA dependent manner (Figure 3), although it remains unknown how the specific interaction between Cka2 and Sir2 is regulated. We mentioned this specific regulation in the text (subsection “Sir2 S473 phosphorylation inhibits DR-mediated lifespan extension by Sir2”).

*3) The data shown in*
Figure 1—figure supplement 3*, that over expression of SIR2 causes resistance to salt and decreases* PMA1 *expression, is an important finding and should be included in the main body of the text/figures*.

As suggested, we moved Figure 1—figure supplement 3 into the main body of the text/figures (Figure 1).

*4) The data regarding Sas2 and Sir2 do not fit your model and are inconsistent amongst themselves. Figure 1–figure supplement 4D shows that the* sir2∆ sas2∆ *double mutant has less H4 K16ac, just like the* sas2∆. *This makes sense. However, the* sir2∆ sas2∆ *double mutant has higher H4K16Ac by ChIP and has higher* PMA1 *expression. The ChIP data suggest additional HATs for H4K16 at these loci; however the WB suggest that Sas2 is responsible for most of H4K16ac inside the cell. You need to explain what is a likely explanation for this contradiction*.

As the reviewers’ comments, the data in Figure 1—figure supplement 3 seem to be inconsistent among themselves. The very low level of H4K16 acetylation shown in Figure 1—figure supplement 3 agrees well with other studies (31; 57). However, Figure 1—figure supplement 3 suggest additional HATs replacing Sas2 in the absence of Sir2, although we do not know which HAT acetylates H4K16 in the *sir2*Δ *sas2*Δ mutant. Actually, it is known that Esa1 (Essential SAS family Acetyltransferase) acts as a secondary HAT for H4K16 acetylation in yeast ([6]; [57]; [56]; [5]; Oppikofer et al., 2011, A dual role of H4H6 acetylation in the establishment of yeast silent chromatin. EMBO J. 30: 2610-2621). We mentioned this possibility in the text (subsection “Sir2 negatively regulates *PMA1* transcription by deacetylating H4K16”).

*5) Your proposal that* PMA1 *and ribosomal protein gene regulation by Sir2 is a novel mechanism of regulating replicative lifespan and mediating DR effects, would be greatly strengthened by testing whether the* sir2-S473 *mutants alter ribosomal DNA recombination/stability. Ideally, you should combine these mutants with a deletion of* FOB1 *to stabilize the rDNA and then test if there is still an effect on lifespan*.

We analyzed the replicative lifespan (RLS) of *fob1*Δ mutant cells expressing *SIR2-S473E* or *SIR2-S473A*. Under 2.0% glucose conditions, the RLS of the *fob1*Δ cells was decreased by the *sir2-S473E* mutation and increased by the *sir2-S473A* mutation. Under 0.5% glucose conditions, however, the *sir2-S473A* allele no longer extended the RLS of the *fob1*Δ cells. These results are similar with those obtained with *FOB1* cells. Thus, we conclude that the effect of phosphorylation at Sir2 S473 on RLS of yeast cells is independent of rDNA recombination/stability. The data are included in Figure 4 and in the text (subsection “Sir2 S473 phosphorylation inhibits DR-mediated lifespan extension by Sir2”).

*6) Does the* S473A *mutant still interact with Cka2-flag through co-IP assays (as in*
Figure 3*)? According to the triple* TPK *mutant result, it shouldn't if phosphorylation at this site is required for the interaction. If it does, there could be other important phosphorylation sites involved. Similarly, they should show that the* S473A *mutant no longer shows the phosphorylation shift on the gel*.

To address the reviewers’ question of whether other phosphorylation sites could be involved in the cAMP-PKA-dependent Sir2 phosphorylation, we analyzed the phosphorylation level of the Sir2-S473A protein in the wild type and *pde2*Δ cells. Although Sir2-S473A proteins were still phosphorylated, the Sir2 phosphorylation induced by cAMP-PKA signaling was completely blocked in Sir2-S473A (Figure 2—figure supplement 2). This result suggests that Sir2 S473 is the major residue for Sir2 phosphorylation through cAMP-PKA signaling. The figure is shown as Figure 1—figure supplement 2 and explained in the text (subsection “cAMP-PKA signaling inhibits Sir2 activity for the transcriptional repression of *PMA1* through serine 473 phosphorylation.”).

*7) The Results and Discussion section is actually all results without any substantial discussion. At a minimum, you need to add an explanation of the proposed model in*
Figure 4*, and relate the new findings and model to other models of Sir2-mediated gene regulation or lifespan effects*.

We separated the Results and Discussion section and added some paragraphs discussing the importance of our new findings in the Sirtuin biology field and the possible connection between Sir2 and organelles involved in cellular aging processes (Discussion).

*8) The actual* P *values for each yeast replicative lifespan assays need to be specifically stated in the figure legends or in a separate table*.

As suggested, we provided the *P* values for each yeast replicative lifespan assay in the figure legends (Figure 4 legend).

*9) For all real-time PCR data (RNA and ChIP), the number of biological replicates* (n) *performed needs to be indicated, to allow the reader to assess the quality and consistency of the data*.

As suggested, we provided the number of biological replicates for all real-time PCR data in the figure legends.

*10) In*
Figure 1—figure supplement 1*, panel B and C, the NaCl sensitivity by* sir2∆ *seems much less than all the other NaCl or monovalent cation sensitivity experiments shown (*Figure 1*;*
Figure 1—figure supplement 1*;,*
Figure 1—figure supplement 2*;*
Figure 1—figure supplement 3*; Figure 1–figure supplement 4A;*
Figure 2*;*
Figure 2—figure supplement 1
*and*
Figure 3*)*. *What is the cause of inconsistency? Are there other biological replicate experiments showing better consistency?*

We agree with the reviewers’ observation that the effect of *sir2*∆ mutation on NaCl sensitivity was more pronounced in our wild type background strain than those in other strains. But, LiCl sensitivity was similar (about 10-fold increase in the *sir2* mutant) in all strains (Figure 1—figure supplement 1). We repeated these experiments so many times and obtained consistent results. More importantly, the level of *PMA1* mRNA in the *sir2* mutant was consistently higher by about 40% than that in the wild type in all strains, and the amount of the Sir2 protein was increased at least two-fold in all strains tested (we added new western data in Figure 1—figure supplement 1). Thus, we think that the seemingly inconsistent results are probably due to strain-specific characteristics related to NaCl sensitivity.

*11) In Figure 1–figure supplement 4, panel A*, *the* sas2∆ *strain grew poorly on YPD, which is even worse than in NaCl. Is this unexpected result reproducible?*

Actually, the growth of the *sas2*∆ strain in YPD is not worse than that in NaCl. To avoid the misunderstanding, we optimized the incubation time after spotting and a new picture is now included in Figure 1—figure supplement 3.